# Impact of a Rice-Centered Diet on the Quality of Sleep in Association with Reduced Oxidative Stress: A Randomized, Open, Parallel-Group Clinical Trial

**DOI:** 10.3390/nu12102926

**Published:** 2020-09-24

**Authors:** Minori Koga, Atsuhito Toyomaki, Yoshinobu Kiso, Ichiro Kusumi

**Affiliations:** 1Department of Psychiatry, Hokkaido University Graduate School of Medicine, Sapporo, Hokkaido 060-8638, Japan; toyomaki@med.hokudai.ac.jp (A.T.); ikusumi@med.hokudai.ac.jp (I.K.); 2Department of Psychiatry, National Defense Medical College, Tokorozawa, Saitama 359-8513, Japan; 3Institute for the Promotion of Business-Regional Collaboration, Hokkaido University, Sapporo, Hokkaido 001-0021, Japan; y-kiso@mcip.hokudai.ac.jp

**Keywords:** rice-centered diet, sleep quality, metabolome, oxidative stress

## Abstract

Previously, we found that a Japanese diet was associated with psychological status, and a combination of rice and miso was related to mental and physical health. We hypothesized that the intake of a rice-based diet affected mental and physical health and aimed to investigate the consequences of a dietary intervention with rice. We conducted a randomized, open-label, parallel-group clinical trial that included 60 participants, who were randomly assigned to receive either rice-based meals or meals with other cereals for three daily meals over 2 months. The participants were surveyed for psychological status and biochemical changes. Sleep quality index scores showed significant improvement after the rice-based intervention. Additionally, blood oxidative stress levels were reduced in the rice-diet group compared with the no-rice-diet group. Although the molecular mechanisms should be investigated in detail, our findings suggest that controlling oxidative stress through the intake of a rice-centered diet may be key to improving sleep quality.

## 1. Introduction

The Japanese diet is popular worldwide and is known for its health benefits. However, few studies have focused on the health benefits of the Japanese diet compared with those of the Mediterranean diet. Rice is a traditional and primary staple food in Japan. The Japanese diet generally comprises rice and other dishes, such as soup, including those with meat, fish, and vegetables. A typical Japanese diet comprises one main dish and two side dishes, so-called “Ichiju-Sansai,” which denotes one soup and three dishes [1]. Thus, in addition to rice, the other two dishes also contribute healthwise to the Japanese diet. A recent study suggested that rice intake improves the quality of sleep [2]. In our previous study, we investigated the association between the intake of a rice-centered diet and brain health [3]. In that study, the intake of traditional Japanese foods such as miso (soybean paste), natto (fermented soybean), and green tea was related to improved depression status, quality of sleep, and degree of impulsiveness. Since that study was etiological, the causal relationships were unclear. Therefore, in the present study, an intervention was conducted using meal instructions concerning staple foods to elucidate whether the intake of a rice-centered diet contributes to brain health. Additionally, to assess the biochemical mechanisms underlying the relationship between these foods and brain health, changes in plasma metabolite levels were determined. The present study provides evidence of the effect of rice or foods consumed with rice on brain health and the underlying biochemical mechanisms.

## 2. Materials and Methods

### 2.1. Participants and Ethics

This study was approved by the institutional review board of the Miyawaki Orthopedic Clinic (Eniwa City, Hokkaido, Japan) prior to study initiation (accession number: 15126) and registered with University Hospital Medical Information Network (UMIN) (UMIN000025723, https://upload.umin.ac.jp/cgi-open-bin/ctr_e/ctr_view.cgi?recptno=R000029579). The study was performed in accordance with the Declaration of Helsinki for Human Research. This intervention study was conducted by the New Drug Research Center. Healthy men and women aged 40–69 years were eligible and provided written informed consent for participation. The inclusion and exclusion criteria are shown in Table 1. The experimental flow is presented in Figure 1. In total, 103 people were initially recruited. In the present study, participant candidates were recruited from the registered monitors at the New Drug Research Center; 41 were excluded due to adherence to the exclusion criteria. Candidate subjects should complete dinner by 21:00 on the day before the pre-examination and then fast (only a small amount of water may be taken). On the day of the examination, the candidate subjects visited the hospital without breakfast and were informed of the contents of this examination according to the consent explanation document, and those who agreed to participate in the examination received the test described below as a pre-examination. After a statistical analysis stratification by age and gender in the pre-examination, all the subjects were assigned a pseudorandom number from 0 to 1. Within all strata, the subjects were assigned to group 1 or 2 in descending order of the pseudorandom number. If it was confirmed that there were significant differences or imbalances in the scores of the Japanese version of the Pittsburgh Sleep Quality Index (PSQI-J), the assignment was repeated until they did not occur. The sample size for the present study was calculated with G*Power 3.2.9 for Windows [4] with an effect size of 0.4 and an alpha error probability of 0.05; the total sample was estimated to be 60 with a statistical power of more than 0.90. According to the result, 62 subjects examined by the investigator were selected as subjects for the main examination. Each group received instructions on staple foods for three daily meals as follows: for group 1, grains other than rice (no-rice-diet group, *n* = 31), and for group 2, rice (rice-diet group, *n* = 31). To avoid withdrawal, rice intake at dinner was allowed in the no-rice-diet group. Each group consumed meals according to the instruction provided for 8 weeks. Two participants withdrew due to personal reasons in the rice-diet group. Finally, 31 participants in the no-rice-diet group and 29 in the rice-diet group were enrolled for further analyses. There was no demographic difference at baseline between the two groups (Table 2).

### 2.2. Questionnaires

Dietary patterns were surveyed using the original questionnaire for dietary patterns and the brief-type self-administered diet history questionnaire (BDHQ) [5]. The BDHQ comprises 75 questions that assess food consumption habits over the past month. Energy and food intakes were calculated using an ad hoc computer algorithm [5]. Dietary patterns were investigated at 0, 4, and 8 weeks after intervention initiation. The value of each item was energy-adjusted by density methods (g/1000 kcal) in each subject.

Depression status was assessed using the Japanese version of the Patient Health Questionnaire-9 (PHQ-9) [6]. This questionnaire was originally established to screen for and diagnose depression with a threshold PHQ-9 score of 10. In addition to the use of criteria-based diagnoses for depressive disorders, the PHQ-9 has been validated as a reliable tool for the measurement of depression severity [7]. PHQ-9 scores correlate with depression on the following scale: none (0 to <5), mild (5 to <10), moderate (10 to <15), moderately severe (15 to <20), and severe (>20).

Anxiety was assessed using the State-Trait Anxiety Inventory (STAI) [8]. The STAI is commonly used for the measurement of trait and state anxiety. This questionnaire comprises 20 items each for the assessment of trait and state anxiety. All items are rated on 4-point scales (from “almost never” to “almost always”). Higher scores indicate greater levels of anxiety.

Sleep quality was measured using the Japanese version of the Pittsburgh Sleep Quality Index (PSQI-J) [9]. The questionnaire comprises seven domains: subjective sleep quality, latency, duration, sleep efficiency, nighttime disturbances, sleep medication use, and daytime dysfunction. The PSQI-J score is the sum of all the subscores from the seven domains. Higher scores reflect more frequent occurrences of sleep disturbances, indicating lower sleep quality.

Scores at each time point were subtracted from the score at week 0 and used for further statistical analysis.

### 2.3. Assessment of Metabolites

A total of 50 µL of plasma was added to 450 µL of methanol-containing internal standards (Solution ID: H3304-1002, Human Metabolome Technologies, Tsuruoka, Japan) at 0 °C for enzyme inactivation. The extract solution was thoroughly mixed with 500 µL of chloroform and 200 µL of Milli-Q water and centrifuged at 2300× *g* at 4 °C for 5 min. A total of 350 µL of the upper aqueous layer was centrifugally filtered through a Millipore 5 kDa cutoff filter for protein removal. The filtrate was centrifugally concentrated and resuspended in 50 µL of Milli-Q water for capillary electrophoresis–mass spectrometry (CE-MS) analysis. The metabolome measurements were performed by a facility service at Human Metabolome Technologies, Tsuruoka, Japan.

Capillary electrophoresis time-of-flight mass spectrometry (CE-TOFMS) was performed using an Agilent CE Capillary Electrophoresis System equipped with an Agilent 6210 Time-of-Flight mass spectrometer, Agilent 1100 isocratic high-performance liquid chromatography pump, Agilent G1603A CE-MS adapter kit, and Agilent G1607A CE-electrospray ionization-MS sprayer kit (Agilent Technologies, Waldbronn, Germany). The systems were controlled by Agilent G2201AA ChemStation software version B.03.01 for CE (Agilent Technologies, Waldbronn, Germany). Metabolites were analyzed using a fused silica capillary (50 μm i.d. × 80 cm total length) with commercial electrophoresis buffer (Solution ID: H3301-1001 for cation analysis and H3302-1021 for anion analysis, Human Metabolome Technologies) as the electrolyte. The sample was injected at a pressure of 50 mbar for 10 s (approximately 10 nL) in the cation analysis and 25 s (approximately 25 nL) in the anion analysis. The spectrometer was scanned from *m/z* 50 to 1000. Other conditions were as described previously [10,11,12]. Peaks were extracted using automatic integration software, MasterHands (Keio University, Tsuruoka, Japan), to obtain peak information, including *m/z*, migration time for CE-TOFMS measurement (MT), and peak area [13]. Signal peaks corresponding to isotopomers, adduct ions, and other product ions of known metabolites were excluded, and the remaining peaks were annotated with putative metabolites from the Human Metabolome Technologies metabolite database based on their MT and *m/z* values as determined by TOFMS. The peak annotation tolerance range was configured at ±0.5 min for MT and ±10 ppm for *m/z*. In addition, peak areas were normalized against those of the internal standards, and the resultant relative area values were further normalized by sample amount.

### 2.4. Statistics

Comparisons between the two groups were performed using Student’s *t*-test in terms of the participants’ physical examination results. These tests were performed by JMP Pro 13.1.0 (SAS Institute Inc., Cary, NC, USA). Repeated measures analysis of variance was used to evaluate group by time in clinical scales and food consumption during the intervention period. Post hoc comparison between the groups in each time point was conducted using Bonferroni’s test. For the assessment of the metabolites, hierarchical cluster analysis and principal component analysis were performed using Human Molecular Technologies Inc.’s proprietary software PeakStat and SampleStat, respectively. The rice-diet group was stratified by the result of the principal component analysis into nonresponder and responder groups in sleep quality, and comparison between before and after the intervention in each group was performed using Welch’s *t*-test.

## 3. Results

### 3.1. Impact on the Psychological Score by Staple Food Type

There were no significant changes in the PHQ-9 and STAI scores in any of the intervention periods—0 to 8 weeks—in the repeated measures analysis of variance. The sleep disturbance score for the evaluation of sleep quality that showed a trend group-by-time interaction was indicated in the PSQI-J score between the two intervention groups (*p* < 0.08, Table 3). A significant interaction between 0 and 4 weeks was indicated (data not shown). The comparison between groups at 4 weeks indicated a significant decrease in the rice-diet group (*p* < 0.05 using Bonferroni’s test).

### 3.2. Impact on Food Intake Patterns by Staple Food Type

Changes in dietary patterns with the instructed intervention in the study participants throughout the intervention period were assessed. Investigation of the food intakes using the BDHQ at weeks 0, 4, and 8 demonstrated the presence of different patterns in the changes in intake of some foods between the no-rice-diet and rice-diet groups (Figure 2 and Appendix A). Intakes of fish with edible bones, low-fat fish, bread, noodles, and rice indicated significant interaction in group by time. Miso intake indicated a trend interaction. Of these foods, intakes of bread and noodles were significantly higher in the no-rice-diet group than in the rice-diet group. Intake of rice was significantly higher in the rice-diet group than in the no-rice-diet group. Post hoc comparison indicated higher levels of bread and noodle consumption and lower level of rice consumption in the no-rice-diet group at weeks 4 and 8. There was no significant change in food intake in both groups except for changes in these cereals’ intake during the 8-week intervention (Figure 2). Although the intake patterns of staple foods and some foods differed in these groups, there were no significant differences between the groups in terms of weight or BMI after the 8-week intervention period. 

### 3.3. Dietary Patterns in the Rice-Diet Group with/without an Effect on Sleep Quality

There were two responses to the intervention within the rice-diet group in which significant sleep quality improvements were noted. Some participants had improved sleep quality, while others did not (based on the cutoff point of 6 defined for sleep disorders in the PSQI-J [9]; in those with a score higher than 6 before the intervention, the score reduced to lower than 6 after it). Regarding participants with a PSQI-J score of 6 or higher at the beginning of the intervention, those with a score lower than 6 after 8 weeks of intervention were classified into the “responder group,” and those whose scores did not fall below 6 after the intervention were grouped into the “nonresponder group.” There was no significant interaction in the anxiety scaled by STAI and depression status scaled by PHQ-9 between the responder and nonresponder groups. The scales of PSQI-J in the nonresponder group at 4 and 8 weeks were significantly lower than those of the responder group (Table 4). All food items did not indicate significant group-by-time interaction except for egg (see Table 5 for the BDHQ items with significance and Appendix A for all items). However, no significant difference was indicated between groups at each time point. Thus, it was considered that dietary patterns in nonresponder and responder groups were significantly different.

### 3.4. Metabolic Changes in the Rice-Diet Group with/without an Effect on Sleep Quality

We conducted biochemical profile analyses in the groups with differences in sleep quality responses due to the intake of rice. Levels of metabolites in the plasma derived from the sleep responder and nonresponder groups after the 8-week intervention were comprehensively analyzed. Consequently, 175 metabolites were detected. The results of the principal component analysis in the overall changes between before and after the intervention in all the samples for these molecules indicated that neither group showed significant changes before and after the intervention. However, the nonresponse group and the response group formed different clusters (Figure 3). Intergroup comparisons of the individual metabolites identified molecules of which the expression levels had changed: 2-oxoisovaleric acid, 5-oxoproline, *cis*-aconitic acid, cysteine glutathione disulfide, cystine, ornithine, *S*-methylcysteine, uric acid, and undefined molecule XA0027 for the responder group, and 3-aminobutyric acid, glutamine, isobutyrylcarnitine, isocitric acid, *N*^5^-ethylglutamine, and sulfotyrosine for the nonresponder group. In addition, some molecules differed in their plasma concentrations before and after the intervention in both groups (undefined molecule XC0120) (see Table 6 for the significant molecules detected in the metabolite assessment and Appendix A for all the detected metabolites).

## 4. Discussion

The correlation between sleep quality and rice consumption has been reported in previous epidemiological studies, including in one we conducted [2,3]. The findings of the present study suggest that rice consumption may be a factor in the improvement of sleep quality. Since the Japanese diet comprises staple foods as well as main and side dishes [1], regarding the foods contributing to the improvement of sleep quality, foods peculiar to rice-centered diets should be considered in addition to rice. Unexpectedly, no food items with group-by-time interaction have been identified in the consumption except for staple foods between the rice-diet group and no-rice-diet group by dietary guidance. Higher levels of bread and noodle consumption in the no-rice-diet group and higher levels of rice consumption in the rice-diet group were shown through the intervention period. These foods are suggested to have contributions to sleep quality. Since noodle consumption has been associated with poor sleep quality [2], the difference in sleep quality between the rice-diet and no-rice-diet groups might be a result of the adverse effects of noodle consumption in the non-rice-eating group.

We also investigated biochemical changes in those who had a rice-centered diet with isocitric acid, 2-oxoisovaleric acid, and aconitic acid, which belong to the energy pathway, the Krebs cycle, which is involved in the circadian rhythm. However, Bell did not observe any strong features that contributed to sleep disappearance in glycolysis, gluconeogenesis, or glycogen metabolism [14]. The changes in the levels of isocitrate, 2-oxovaleric acid, and aconitinic acid in responders and nonresponders were inconsistent.

While few reports have focused on isobutyrylcarnitine by itself, a previous report demonstrated that carnitine—its liberated form—affected circadian rhythms through lipid metabolism and contributed to correcting a disturbed circadian rhythm [15]. As the levels of isobutyrylcarnitine decreased significantly in the nonresponders, the effect of a rice diet may have been canceled by the disruption of the circadian rhythm. Improvement of sleep quality by ornithine intake has been demonstrated in human studies [16,17].

The metabolite 5-oxoproline is a precursor of glutathione in the γ-glutamyl cycle [18], and that cysteine glutathione disulfide is a glutathione delivery agent [19]. Cystine is an oxidized form of cysteine; it is a thiol group donor to glutathione, which is reduced in vivo and acts as an antioxidant [20] and is also a precursor in the γ-glutamyl cycle. *S*-methylcysteine was suggested to have a role of antioxidant [21]. These antioxidant agents showed an increase in the responder group. The levels of *N*^5^-ethylglutamine, a theanine, which is known as an antioxidant [22], and the reduction in theanine levels in the nonresponder group in this study were indicated. Uric acid aids in reactive oxygen species (ROS) production and is known to cause oxidative stress [23], and the levels of uric acid were significantly decreased in the responder group, suggesting that resistance to oxidative stress is associated with improvements in sleep quality. Interestingly, about half of the identified candidate molecules are related to oxidative stress. Recent studies indicate that abnormal oxidative stress is an element constituting sleep disturbance [24] and the pathology of mental disorders, such as schizophrenia, bipolar disorder, and depression [25,26,27]. Several studies have reported that abnormal oxidative stress levels disturb the circadian rhythm and lead to sleep quality deterioration (see review [28]). In the metabolome analysis in the present study, the decrease in the levels of molecules involved in elevated oxidative stress and the increase in the levels of molecules involved in increased antioxidant levels in the responder group support this mechanism. We could not conclude a relationship of the changes of 3-aminobutyric, sulfotyrosine, XA0027, and XC0120 (these have not been identified yet) with an improvement of sleep quality.

There are some limitations to the present study. First, the sample size was small to explain the association of rice-centered diet with sleep quality. Additional sample numbers with more statistical power would be needed to validate the results in the present study. Second, because we were also interested in foods consumed together due to changes in staple foods, the intervention instructions only mentioned staple foods, and thus there were differences in the intake of multiple foods between the rice and nonrice food groups. Therefore, we were unable to identify foods that directly affected sleep quality. Further identification should be conducted. Third, in this study, it is impossible to determine whether the improvement of sleep quality in the rice-diet group was positively influenced by the intake of rice and related foods or negatively influenced by the noodle intake in the non-rice-diet group.

In conclusion, the present study demonstrated that the intake of a rice-centered diet may have a significant effect on sleep quality through an intervention. As far as we know, there have been no previous studies of rice-based dietary interventions. The results of the present study support the results of our and other groups’ previous epidemiological studies. However, the effect on sleep quality was constitutional; thus, we observed both responders and nonresponders in the rice-diet group. Nonetheless, we identified some metabolites specifically related to sleep quality improvements, and a number of them were associated with oxidative stress; therefore, the intake of a rice-centered diet may reduce oxidative stress. Since oxidative stress is estimated to be a negative risk factor for not only sleep disturbance but also mental disorders, the intake of a rice-centered diet may prove effective in improving brain health.

## Figures and Tables

**Figure 1 nutrients-12-02926-f001:**
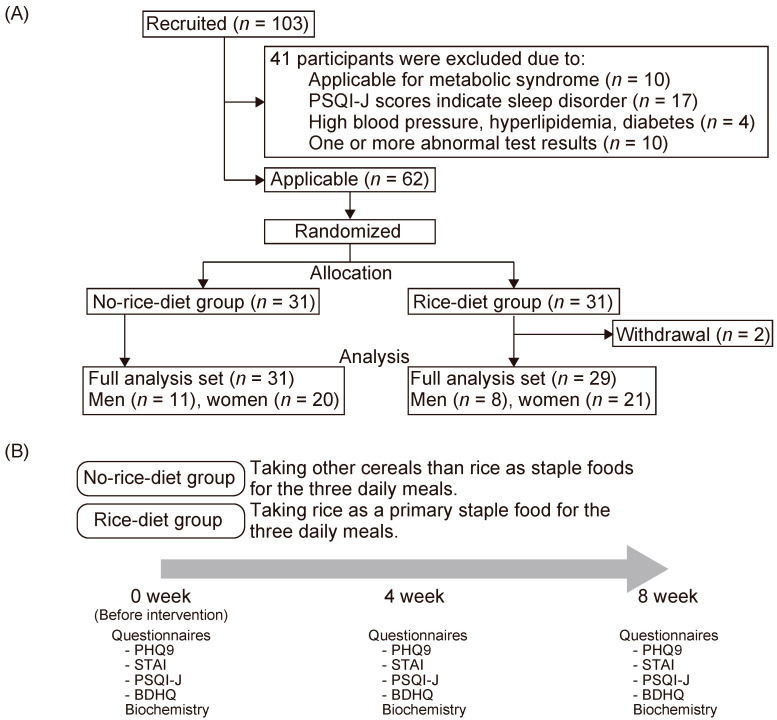
Recruitment and study design. (**A**) A study flowchart. A total of 103 candidates were recruited, but only 62 were eligible. They were assigned to two groups: other cereals group and rice group. Two participants withdrew, while 31 candidates (11 men and 20 women) were assigned to the other cereals group, and 29 candidates (8 men and 21 women) were assigned to the rice group for analysis in the present study. (**B**) An intervention and analysis flowchart. Other cereals than rice were assigned to the other cereals group as staple foods for the three daily meals, and rice as a primary staple food was assigned to the rice group for the three daily meals for 8 weeks. All participants were surveyed using psychological assessment questionnaires: Patient Health Questionnaire-9 (PHQ-9), State-Trait Anxiety Inventory (STAI), and Japanese version of Pittsburgh Sleep Quality Index (PSQI-J), and a dietary pattern assessment questionnaire, brief-type self-administered diet history questionnaire (BDHQ). Their blood was also taken to assess their metabolites.

**Figure 2 nutrients-12-02926-f002:**
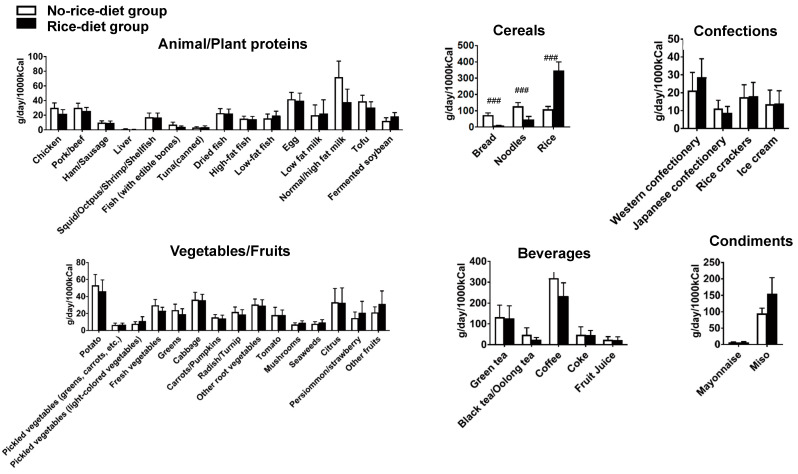
The amount of food intake at week 8 in the test groups. The mount of food intake at the end of the intervention period calculated based on BDHQ responses (*n* = 31 no-rice-diet group and *n* = 29 rice-diet group). Data are means ± 95% confidence intervals. Comparison between no-rice-diet and rice-diet groups after repeated measures analysis of variance was performed (Bonferroni’s correction, ### *p* < 0.001).

**Figure 3 nutrients-12-02926-f003:**
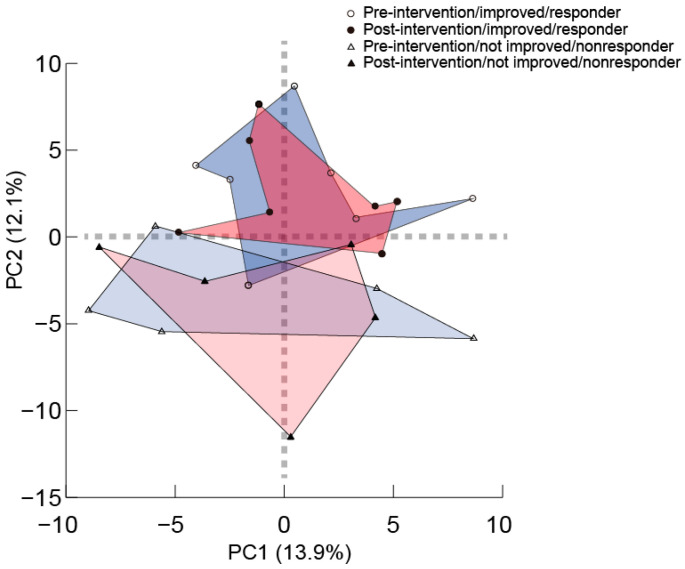
The result of principal component analysis for changes of metabolites between before and after intervention in the rice group. Circles indicate participants who were improved in sleep quality by the intervention (*n* = 5 nonresponder and *n* = 7 responder). Triangles indicate participants who were not improved in sleep quality by the intervention. Open marks indicate before intervention, and solid marks indicate after intervention. PC1 and PC2 show primary and secondary principal components, respectively, and the values in the parentheses indicate contribution for each principal component.

**Table 1 nutrients-12-02926-t001:** Inclusion and exclusion criteria in this study.

Inclusion
A score of 4 or higher in the Japanese version of the Pittsburgh Sleep Quality Index
Match one or more items in the diagnostic criteria for metabolic syndrome
Waist circumference: men, ≥85 cm; women, ≥90 cm
Triglyceride level: >150 mg/dL
High-density lipoprotein cholesterol level
Regularly eating three meals per day
Eating staple food other than rice more than once in daily meals
**Exclusion**
Currently taking glucose metabolism, lipid metabolism, and blood pressure-improving drugs
Receiving treatment for insomnia or sleep disorders
Systolic blood pressure: <90 mmHg
Pregnant women or women who want to conceive

**Table 2 nutrients-12-02926-t002:** Physical examination results in the participants.

Variables	No-Rice-Diet Group	Rice-Diet Group	*p*-Value
Number of subjects	31	29	-
Sex (male:female)	11:20	8:21	0.51
Age (years, mean ± SD)	55.35 ± 8.04	55.79 ± 7.42	0.83
Height (cm, mean ± SD)	159.89 ± 8.78	159.82 ± 8.39	0.98
Weight (kg, mean ± SD)	67.72 ± 9.34	65.62 ± 8.93	0.38
BMI (mean ± SD)	26.42 ± 2.19	25.61 ± 2.01	0.14
Body temperature (°C, mean ± SD)	36.23 ± 0.37	36.18 ± 0.45	0.64
Waist (cm, mean ± SD)	91.36 ± 6.43	90.99 ± 6.20	0.82
Systolic blood pressure (mmHg, mean ± SD)	128.60 ± 19.31	130.16 ± 2.81	0.73
Diastolic blood pressure (mmHg, mean ± SD)	80.94 ± 12.68	81.24 ± 11.19	0.92

*p*-Values were calculated by chi-square test for sex and by *t*-test for other items. Abbreviations: SD, standard deviation; BMI, body mass index.

**Table 3 nutrients-12-02926-t003:** Changes in psychological indices and sleep quality during intervention.

		Duration of Intervention		*p*-Values *
	Intervention Group	4 Weeks	8 Weeks	Group-by-Time Interaction
PHQ-9	No-rice-diet	−0.1	−0.3	0.51
		(−1.1–0.8)	(−1.4–0.8)	
	Rice-diet	−0.9	−0.8	
		(−2.0–0.2)	(−1.9–0.2)	
STAI state	No-rice-diet	−1.2	−2.3	0.41
		(−3.4–1.0)	(−4.3–−0.2)	
	Rice-diet	−3.7	−4.2	
		(−6.9–−0.5)	(−7.5–−0.9)	
STAI trait	No-rice-diet	1.2	0.9	0.62
		(0.1–2.2)	(−0.4–2.2)	
	Rice-diet	0.3	0.4	
		(−1.3–1.8)	(−1.2–2.0)	
PSQI-J	No-rice-diet	0.0	−0.4	0.08 ^+^
		(−0.6–0.6)	(−1.2–0.3)	
	Rice-diet	−1.0 ^#^	−1.2	
		(−1.7–−0.4)	(−1.9–−0.6)	

Values are means (95% confidence interval) of subtracted scores at week 0 (*n* = 31 no-rice-diet group and *n* = 29 rice-diet group); * *p*-values were calculated by repeated measures analysis of variance to investigate interaction in group by time (week periods of the intervention) (^+^, trend *p* < 0.1). Comparison of the difference in clinical scales between no-rice-diet and rice-diet groups at each time point was conducted using Bonferroni’s test (^#^, *p* < 0.05). Abbreviations: PHQ, Patient Health Questionnaire; STAI, State-Trait Anxiety Inventory; PSQI-J, Japanese version of the Pittsburgh Sleep Quality Index.

**Table 4 nutrients-12-02926-t004:** Changes in psychological indices and sleep quality during intervention in the rice-diet group with/without effect on sleep quality.

		Duration of Intervention		*p*-Values *
	Intervention Group	4 Weeks	8 Weeks	Group-by-Time Interaction
PHQ-9	Nonresponder	0.0	0.0	0.38
		(−1.5–1.5)	(−2.8–2.8)	
	Responder	−2.7	−2.6	
		(−6.2–0.8)	(−6.1–0.9)	
STAI state	Nonresponder	−5.4	−8.0	0.61
		(−19.5–8.7)	(−19.2–3.2)	
	Responder	−1.4	−2.9	
		(−8.2–5.4)	(−10.2–4.5)	
STAI trait	Nonresponder	1.0	−1.2	0.55
		(−4.0–6.0)	(−5.5–3.1)	
	Responder	1.7	1.3	
		(−2.9–6.3)	(−3.8–6.4)	
PSQI-J	Nonresponder	0.2	−0.2	0.0064
		(−1.6–2.0)	(−1.6–1.2)	
	Responder	−3.3 ^##^	−3.4 ^##^	
		(−4.6–−2.0)	(−4.8–−2.0)	

Values are means (95% confidence interval) of subtracted scores at week 0 (*n* = 5 nonresponder and *n* = 7 responder). * *p*-Values were calculated by repeated measures analysis of variance to investigate interaction in group by time (week periods of the intervention). Post hoc multiple comparisons between responder and nonresponder groups at each time point were conducted using Bonferroni’s test (^##^, *p* < 0.01). Abbreviations: PHQ, Patient Health Questionnaire; STAI, State-Trait Anxiety Inventory; PSQI-J, Japanese version of the Pittsburgh Sleep Quality Index.

**Table 5 nutrients-12-02926-t005:** Changes in food intake during intervention in the rice-diet group with/without effect on sleep quality.

		Duration of Intervention (g/day/1000 kcal)			*p*-Values *
Food Item in BDHQ	Group	0 Weeks	4 Weeks	8 Weeks	Group-by-Time Interaction
Eggs	Nonresponder	29	33.1	32.0	0.013
		(9.5–48.5)	(13.8–52.4)	(13.4–51)	
	Responder	25	40.5	39.6	
		(8.6–41.4)	(14.5–66.6)	(18.1–61.2)	

Values are means ± standard deviation (95% confidence interval) (*n* = 5 nonresponder and *n* = 7 responder). Abbreviation: BDHQ, brief-type self-administered diet history questionnaire. * *p*-Values were calculated by repeated measures analysis of variance to investigate interaction in group by time (week periods of the intervention). Post hoc multiple comparisons between responder and nonresponder groups at each time point were conducted using Bonferroni’s test.

**Table 6 nutrients-12-02926-t006:** Identified metabolites among the test groups.

Comparative Analysis					
ID	Compound Name	Post-Intervention vs. Pre-Intervention (Responder), *n* = 7		Post-Intervention vs. Pre-Intervention (Nonresponder), *n* = 5	
		Fold Change	*p*-Value	Fold Change	*p*-Value
A_0010	2–Oxoisovaleric acid	**0.9**	0.036	1.0	0.588
C_0014	3-Aminobutyric acid	0.9	0.445	**1.1**	**0.002**
A_0017	5-Oxoproline	**1.2**	**0.011**	1.1	0.068
A_0042	*cis*-Aconitic acid	**1.2**	**0.027**	1.2	0.051
C_0123	Cysteine glutathione Disulfide	**2.6**	**0.002**	1.9	N.A
C_0106	Cystine	**2.6**	**0.005**	1.9	0.122
C_0058	Glutamine	1.1	0.076	**1.1**	**0.021**
C_0103	Isobutyrylcarnitine	0.8	0.188	**0.6**	**0.043**
A_0050	Isocitric acid	1.1	0.228	**1.3**	**2.8 × 10^−4^**
C_0077	*N*^5^-Ethylglutamine	0.9	0.504	**0.6**	**0.038**
C_0044	Ornithine	**1.3**	**0.022**	1.3	0.052
C_0047	*S*-Methylcysteine	**1.8**	**0.013**	1.5	0.102
A_0070	Sulfotyrosine	1.0	0.739	**1.2**	**0.004**
A_0036	Uric acid	**0.9**	**0.022**	1.0	0.450
A_0062	XA0027	**0.8**	**0.034**	1.1	0.816
C_0118	XC0120	**2.9**	**8.1 × 10^−4^**	**2.4**	**0.036**

Comparisons between pre- and post-intervention in the responder and nonresponder groups were performed by Welch’s *t*-tests (*n* = 5 nonresponder and *n* = 7 responder). Bold letters indicate statistically significant difference (*p* < 0.05).

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
