# Peer review of "Impact of a Rice-Centered Diet on the Quality of Sleep in Association with Reduced Oxidative Stress: A Randomized, Open, Parallel-Group Clinical Trial"

_nutrients, 2020, doi:10.3390/nu12102926_

Round 1
Reviewer 1 Report
Using a randomized clinical trial, the authors aimed to test the effects of rice on sleep. The research question is interesting. Overall the study is well designed. However, there are some major concerns.
Major:
- There is no mention of the strength and limitation of the study. Whether the findings are generalizable is unknown. In a culture with rice as a staple food, people who did not consume rice could be different in other behaviours or have some underline conditions. Rice intake was associated with a low BMI in some Asian countries. The study included participants who are overweight/obese. Is there any change in body weight during the follow-up? Although body weight is not the main research outcome, it is interesting to know.
- The sample size may be underpowered in the comparison of the metabolites. Because multiple comparisons were made, some significant p values may be purely due to chance.
- The presentation of the difference in the food intake is also suffering from the small sample size with multiple comparisons. It is difficult to get a whole picture of how different the dietary patterns are. The authors could try to construct over dietary patterns using factor analysis.
Minor:
- Table 4: what are the units of the food intake?
- Table 5: sample size should be provided.
- Did the authors test the metabolites in the no-rice group?
- The discussion needs to be concise.
Author Response
Response to Reviewer 1:
Thank you for reviewing our manuscript. We have realized the values of food consumption were raw values. the value of each item should be energy-adjusted by density methods (g/1,000 kcal). We have corrected all values derived from BDHQ questionnaire in this revision. One of the authors pointed to analyze subtracted values in clinical scales from those at 0w. There is no change in significance, but the numbers in the table have been changed in this revision.
Major:
- There is no mention of the strength and limitation of the study. Whether the findings are generalizable is unknown. In a culture with rice as a staple food, people who did not consume rice could be different in other behaviours or have some underline conditions. Rice intake was associated with a low BMI in some Asian countries. The study included participants who are overweight/obese. Is there any change in body weight during the follow-up? Although body weight is not the main research outcome, it is interesting to know.
We have added a paragraph describing the limitation of the present study. As far as we know, this paper is first study to investigate causality of rice-centered diet on sleep quality. We have this sentence in the conclusion.
It is interesting to investigate differences in behaviors in those who take rice much or do not. However, the aim of the present study is to investigate effect of rice-centered diet on mental health. At the beginning of the intervention period, there were no differences in bread and rice consumption in both of test groups.
We also tested the body weight and BMI, and these indicated no significant changes before and after the 8 weeks intervention. The information has been added in the result section, part 3.2.
- The sample size may be underpowered in the comparison of the metabolites. Because multiple comparisons were made, some significant p values may be purely due to chance.
We have mentioned about small sample size as a limitation on the present study in the discussion session. Koike et al reported results of metabolome analysis with multiple comparison without analysis of variance (S Koike et al, PMID: 24713860). However, as you mentioned, the results may include some items identified by chance. The metabolome investigation is as a first screening to explore metabolites related to sleep quality. Further investigation should be conducted with more sample size and strict statistics.
- The presentation of the difference in the food intake is also suffering from the small sample size with multiple comparisons. It is difficult to get a whole picture of how different the dietary patterns are. The authors could try to construct over dietary patterns using factor analysis.
We agree your point. The approach of identifying dietary patterns through factor analysis is important way. The Our team previously investigated dietary patterns conducted with factor analysis (A Toyomaki et al., PMID: 28704469). However, in the present study, we aimed to explore individual foods influenced by rice-centered diet. Thus, we tested each item in no-rice and rice-groups. We have also mentioned about small sample size in the limitation section in the discussion.
Minor:
Table 4: what are the units of the food intake?
Units have been added in the tables.
Table 5: sample size should be provided.
Sample numbers have been indicated in each table.
Did the authors test the metabolites in the no-rice group?
We have not investigate no-rice group in this study. Since there seemed 2 clusters of sleep quality in the rice-diet group, we decided to use them so that we can find
The discussion needs to be concise.
We deleted some of description in the discussion part for metabolome analysis.
Reviewer 2 Report
This study tried to investigate the effect of rice or foods consumed with rice on sleep and brain health. However, there are a couple of things are not clear to me.
- study design-the participants were randomly assigned to rice-based group and non-rice based group. As the authors claimed that meals other than staples (rice or non-rice) may contribute to the sleep quality, but how they controlled the meals for each group is not clear. The authors wrote that " To avoid withdrawal, the staple dinner food was not strictly enforced in the no-rice diet group. Each group consumed meals according to the instruction provided for 8 weeks. " What do they mean by "to avoid withdrawal"? Did they mean that the meals accompanied with rice or non-rice is not restricted? but they also said these meals were provided according to the instructions, what are these instructions? The study groups seemed to compare between rice based and non-rice based in terms of sleep outcomes, but how can you compare if you don't control the factors other than the rice or non-rice (i.e. the meals)?
- there were significant differences of the meals between rice-based group and non-rice based groups. How to differentiate the net effect of rice on the outcome?
- Discussion- I think the authors have discussed many other studies and possible mechanism rather than their present study. More discussions are needed for their own study design, and results etc.
Author Response
Response to Reviewer 2:
Thank you for reviewing our manuscript. We have realized the values of food consumption were raw values. the value of each item should be energy-adjusted by density methods (g/1,000 kcal). We have corrected all values derived from BDHQ questionnaire in this revision. One of the authors pointed to analyze subtracted values in clinical scales from those at 0w. There is no change in significance, but the numbers in the table have been changed in this revision.
- Study design-the participants were randomly assigned to rice-based group and non-rice based group. As the authors claimed that meals other than staples (rice or non-rice) may contribute to the sleep quality, but how they controlled the meals for each group is not clear. The authors wrote that " To avoid withdrawal, the staple dinner food was not strictly enforced in the no-rice diet group. Each group consumed meals according to the instruction provided for 8 weeks. " What do they mean by "to avoid withdrawal"? Did they mean that the meals accompanied with rice or non-rice is not restricted? but they also said these meals were provided according to the instructions, what are these instructions? The study groups seemed to compare between rice based and non-rice based in terms of sleep outcomes, but how can you compare if you don't control the factors other than the rice or non-rice (i.e. the meals)?
We hypothesized rice-centered diet improves mental health via food taken together with rice (Koga et al., PMID 28968452). Our aim is to investigate whether rice-centered diet, but not only rice, contributes to improve mental health. Therefore, multiple food items that may contribute to improve sleep quality have been identified. Further identification should be done. We have described this in the limitation section in the discussion.
In this study, we instructed three daily meals with no-rice-diet and rice-centered diet. However, In Japanese people, it was expected that a diet completely devoid of rice would result in a large number of withdrawals. To avoid withdrawal, rice intake at dinner was allowed in the no-rice diet group. By this instruction, the rice intake is significantly difference between no-rice-diet and rice-diet group (Table 4). We thus expected the effect of rice-centered diet was be able to be investigated. We have re-edited method section explaining group assignment.
- There were significant differences of the meals between rice-based group and non-rice based groups. How to differentiate the net effect of rice on the outcome?
As mentioned above, our aim is to investigate effects of rice-centered diet on mental health. However, we cannot evaluate foods individually. We mentioned about this in the limitation section.
- Discussion- I think the authors have discussed many other studies and possible mechanism rather than their present study. More discussions are needed for their own study design, and results etc.
It seemed too much detail in description about metabolome analysis. Therefore, some descriptions have been removed. Limitation and findings from the present study have been added.
Round 2
Reviewer 1 Report
The responses to my comments are satisfactory.
Author Response
Thank you for taking time for reviewing our manuscript.
Reviewer 2 Report
Thanks the authors for the improvement. However, the major issue is that they cannot able to differentiate foods accompanied with rice/non-rice, although they claimed this in the limitation.
Author Response
Response to Reviewer 2
- Thanks the authors for the improvement. However, the major issue is that they cannot able to differentiate foods accompanied with rice/non-rice, although they claimed this in the limitation.
We have added a figure so that changes in food intake impacted by the instruction of the intervention can be observed (Figure 2). Cereals intake indicated significant changes during the intervention period, however, other items didn't show a significant difference between No-rice-diet and Rice-diet groups.
